# Remote Sensing Estimation of CDOM and DOC with the Environmental Implications for Lake Khanka

**Sining Qiang** [1,2], **Kaishan Song** [2], **Yingxin Shang** [2], **Fengfa Lai** [2], **Zhidan Wen** [2], **Ge Liu** [2], **Hui Tao** [2] and **Yunfeng Lyu** [1,*]

1   School of Geographic Science, Changchun Normal University, Changchun 130102, China; qiangsining@iga.ac.cn
2   Northeast Institute of Geography and Agroecology, Chinese Academy of Sciences, Changchun 130102, China; songkaishan@iga.ac.cn (K.S.); shangyingxin@iga.ac.cn (Y.S.); laifengfa@iga.ac.cn (F.L.); wenzhidan@iga.ac.cn (Z.W.); liuge@iga.ac.cn (G.L.); taohui@iga.ac.cn (H.T.)
*   Correspondence: lvyunfeng@ccsfu.edu.cn

**Abstract:** Chromophoric dissolved organic matter (CDOM) is a significant contributor to the biogeochemical cycle and energy dynamics within aquatic ecosystems. Hence, the implementation of a systematic and comprehensive monitoring and governance framework for the CDOM in inland waters holds significant importance. This study conducted the retrieval of CDOM in Lake Khanka. Specifically, we use the GBDT ($R^2 = 0.84$) algorithm which performed best in retrieving CDOM levels and an empirical relationship based on the situ data between CDOM and dissolved organic carbon (DOC) to indicate the distribution of DOC indirectly. The performance of the CDOM-DOC retrieval scheme was reasonably good, achieving an $R^2$ value of 0.69. The empirical algorithms were utilized for the analysis of Sentinel-3 datasets from the period 2016 to 2020 in Lake Khanka. The potential factors that contributed to the sources of DOM were also analyzed with the humification index (HIX). The significant relationship between CDOM and DOC (HIX and chemical oxygen demand (COD)) indicated the potential remote sensing application of water quality monitoring for water management. An analysis of our findings suggests that the water quality of the Great Khanka is superior to that of the Small Khanka. Moreover, the distribution of diverse organic matter exhibits a pattern where concentrations are generally higher along the shoreline compared to the center of the lake. Efficient measures should be promptly implemented to safeguard the water resources in international boundary lakes such as Lake Khanka and comprehensive monitoring systems including DOM distribution, DOM sources, and water quality management would be essential for water resource protection and government management.

**Keywords:** chromophoric dissolved organic matter; dissolved organic carbon; chemical oxygen demand; water quality; remote sensing

## 1. Introduction

Lakes play a significant role within the terrestrial hydrosphere. In addition to serving as the foundation for sustaining human life and supporting production activities, they also play a crucial role in hydrological regulation and the preservation of ecosystem functions. The lake ecosystem serves as a dependable medium for the storage, transformation, and transportation of carbon within inland areas [1]. Dissolved organic carbon (DOC) is an important component of the global organic carbon pool as well as a major component of dissolved organic matter (DOM) [2], and DOC in inland waters plays an important role in the global carbon cycle. Chromophoric dissolved organic matter (CDOM) refers to the light-sensitive component of DOM. CDOM is a prominent optically active compound found in water. It exhibits notable properties such as a high absorption of ultraviolet and visible blue light. The strong absorption of this substance has a notable impact on the optical properties

of the water's surface. However, CDOM serves to reduce the transmission and absorption of ultraviolet and blue light in aquatic environments [3]. The primary function of the spread is to mitigate the detrimental effects of UV-B radiation on aquatic organisms [4]. Additionally, it contributes to safeguarding aquatic ecosystems to a certain degree within the water column. As the colored part of DOM, CDOM is prone to photochemical degradation and microbial decomposition in the natural environment, releasing greenhouse gases such as carbon dioxide and methane to exacerbate global warming. The source, composition, degradation and mineralization of CDOM are related to the global carbon cycle [5].

The investigation of the source and composition of CDOM can provide insights into the movement and alteration of organic pollutants, as CDOM possesses the capacity to adsorb metal ions and organic pollutants [6]. The absorption coefficient of CDOM is considered as a potential measure of dissolved organic carbon, which can help estimate the release of carbon dioxide and assess carbon pools in carbon cycle studies at the same time [7]. CDOM assumes a significant optical role within DOC. By examining the optical properties of CDOM, one can effectively assess the distribution and variations of CDOM within inland waters. This, in turn, serves as an indicator for the distribution and movement of DOC [8]. Additionally, CDOM contributes to the global carbon cycle. As the components of CDOM are complex and easy to degrade, the absorption coefficient at a specific wavelength is usually used as an indicator of its concentration [9,10], and a(350) is usually used to represent the concentration of CDOM and the biogeochemical cycle processes [11,12]. The 3D-EEMs (3Dimensions Excitation-Emission Matrices) were considered as a sensitive way to evaluate the sources and composition of DOM rapidly [13]. The fluorescence index can reflect the source and component characteristics of DOM. HIX semi-quantitatively represents the degree of humification of DOM and the possible duration of organic matter in the environment. Some scholars have found an obvious positive correlation between DOC concentration and CDOM in inland waters [8,10,14,15]. DOC is a reliable regional proxy indicator in these studies [16–19]. The water quality status can be inferred indirectly by examining the relationship between CDOM and its composition parameters and water environment indicators [20–22]. Organic pollutants are also known as oxygen-consuming organic matter, such as chemical oxygen demand (COD) which indicates the amount of oxygen required for the oxidation of organic matter in lakes, which is related to DOM to some extent.

Remote sensing technology is a highly efficient tool for generating maps that depict the distribution of CDOM with both high temporal and spatial resolution [10,14,15]. Studies have shown [23,24] that adding 600 nm to the CDOM inversion model can significantly improve the accuracy of CDOM estimation in turbid waters, and using the band search at 800 nm may indirectly prompt CDOM. Increasing the number of bands used as variables is expected to enhance the performance of the model, resulting in improved fitting results and increased robustness. Zhang [20] found that in the main organic pollutant index of Lake Taihu, there was a significant correlation between COD and CDOM uptake. In their study, Huang et al. [25] were able to effectively obtain the spatial distribution of CDOM and COD in the Liaodong Bay region. This was achieved by utilizing the H J-1/CCD environmental satellite. The establishment of this empirical model has promoted the application of water color remote sensing in the monitoring of water environmental pollution components. Wu et al. [26] evaluated seasonal water quality in Lake Taihu by measuring water quality parameters, such as water temperature, dissolved oxygen (DO), turbidity (Tur), COD, total phosphorus (TP), total nitrogen (TN), and assigning weights to them to evaluate the seasonal water quality of Lake Taihu. Sentinel-3 ocean and land color imager (OLCI) is equipped with a full resolution capability of 300 m for earth observation. Additionally, the revisit period for this instrument is approximately 1.5 days. This dataset contains atmospheric radiation in 21 spectral bands, so Sentinel-3 has the advantage of a short revisit period, and the spatial resolution is suitable for the breadth of the 4380 $km^2$ water area of Lake Khanka. As is known, atmospheric correction methods that are designed for either ocean color or land applications often result in low quality or

even no surface reflectance data for coastal and inland waters. Machine learning algorithms have experienced significant advancements and widespread adoption in recent years, and GBDT (Gradient Boosting Decision Tree) is a kind of machine learning algorithm. It is an integrated learning method based on the CART decision tree. The algorithm integrates and iterates multiple tree models obtained through learning and generalization [27,28]; therefore, machine learning algorithms have been widely used in water color remote sensing research in recent years [29–31]. We developed and validated a novel approach to determine long term CDOM records for Lake Khanka using OLCI unsaturated spectral bands following Rayleigh scattering correction in the present study.

Lake Khanka serves as the official boundary between the nations of China and Russia. Water plays a crucial and indispensable role in various domains [32], including the provision of drinking water, support for industrial and agricultural activities, facilitation of waterway transportation, regulation of climate patterns, and preservation of biodiversity. In recent years, the water environment of Lake Khanka has been affected by global climate change and human activities. Hence, the exploration of the CDOM-DOC inversion model holds immense importance in the context of monitoring and safeguarding the water environment in Lake Khanka. The presence of a significant quantity of organic matter and an excessive amount of nutrients in inland water are poised to result in the discoloration and unpleasant odor of the water body. This phenomenon is known as water pollution, which not only disrupts the regional carbon cycle mechanism essential for maintaining water ecological balance but also poses a threat to human survival and development. The primary aim of this study is to create a remote sensing model for the absorption coefficient of CDOM to establish the linkage for understanding the sources and distribution of DOM. This model will be used to assess the potential for monitoring the water quality of Lake Khanka and will even provide further scientific support and a theoretical basis for protection and governance through remote sensing techniques. Additionally, the study seeks to enhance our understanding of the dynamics between CDOM and DOC to offer a comprehensive and interdisciplinary perspective on CDOM in inland waters.

## 2. Materials and Methods

### 2.1. Study Area

Lake Khanka serves as an international boundary lake, spanning across both China and Russia. The lake under consideration is the largest freshwater lake located within the Ussuri River Basin. Furthermore, Lake Khanka has an open water surface and is divided into two parts by a sand embankment. Small Lake Khanka covers an area of about 176 km$^2$, all of which are located in China, and Great Khanka covers an area of about 4380 km$^2$, located in China and Russia, as shown in Figure 1f,g. The western region of Lake Khanka is characterized by the presence of several rivers, including the Muling River, Lev River, Bailing River, Xintu River, Takhrezh River, and Beyqi River. Additionally, there are over 20 small- and medium-sized rivers, such as the Men River. On the eastern side, the Songacha River serves as the sole outlet for excess water from the lake. As the largest boundary lake in China, Lake Khanka, with its huge volume, has multi-directional functions such as water supply in the two basins, the regulation of river runoff, irrigation for agricultural production, and the maintenance of biodiversity protection ecosystem balance.

### 2.2. Field Sampling Data

The in situ data samples were collected during field expeditions conducted in September 2017, July 2018, October 2020, and September and October 2021, respectively. In the field, a total of 170 surface water samples were collected at depths ranging from 0 to 50 cm. The spatial distribution of the sampling points can be observed in Figure 1a–e. This distribution includes 99 water samples from Small Khanka and 71 water samples from Great Khanka. We used a polyethylene container with a capacity of 2 L to seal and store water away from light and used a mixed fiber filter membrane (pore diameter 0.45 μm) to complete the preliminary filtration of the water sample within 12 h. The water samples

were stored in a refrigerator for further measurement in the laboratory of CDOM absorption, DOC concentration, and other water quality parameters. Additionally, we recorded the latitude and longitude coordinates, altitude, wind speed, atmospheric pressure, and weather conditions at the sampling point.

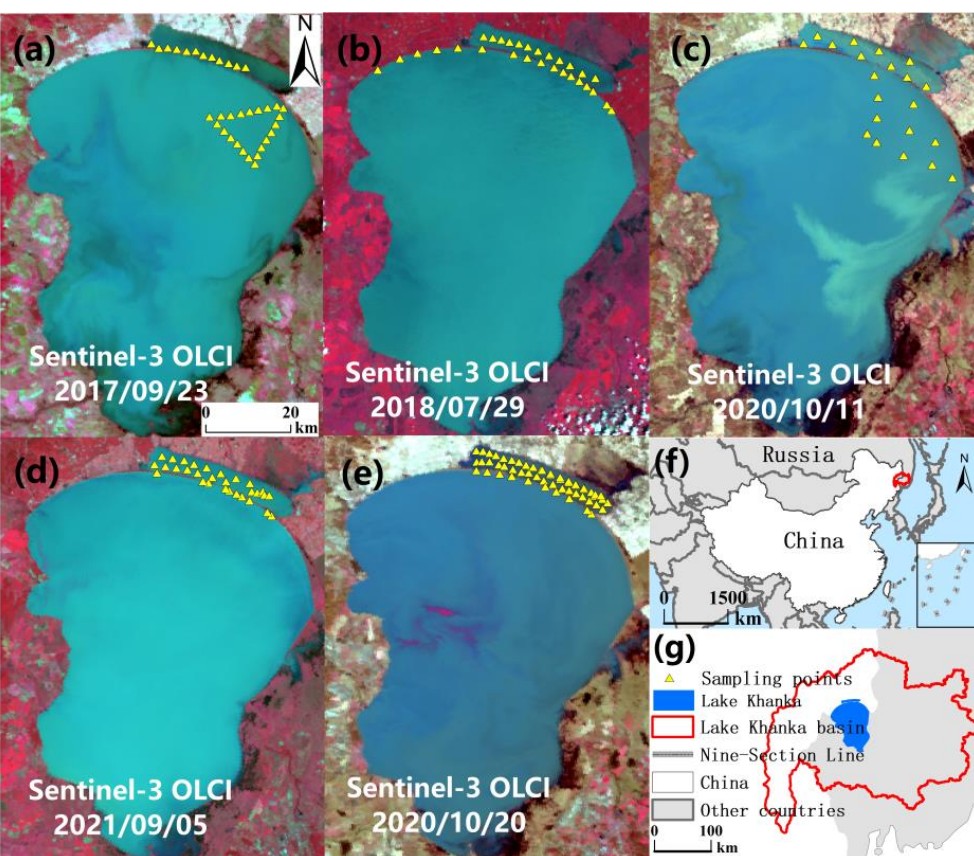

**Figure 1.** (**a–e**) Pseudo-color displays Sentinel-3 OLCI images showing the landscape and sample point locations of the study area, (**f,g**) Lake Khanka, and surrounding drainage basin in the world.

### 2.3. Measurement for CDOM Properties and Water Quality Parameters

The water samples collected in the field were initially filtered using a polycarbonate filter membrane with a pore size of 0.22 μm. This filtration process was performed to obtain the samples specifically for CDOM testing. Subsequently, a Shimadzu UV-spectrophotometer (UV-6200) was utilized to measure the absorbance spectrum in the range of 200–800 nm. Ultrapure water was used as the reference for this measurement. The DOC concentration is obtained by testing the water sample filtered by the Whatman GF/F glass fiber membrane with a pore size of 0.45 μm through the Shimadzu Total Organic Carbon Analyzer (TOC-VCPN). The calculation formula for the CDOM absorption coefficient is as follows:

$$a_{CDOM}(\lambda) = \frac{2.303A(\lambda)}{L} \tag{1}$$

In Equation (1), A($\lambda$) is the absorbance at $\lambda$ nm, and the optical path $L$ is 1 cm. The absorption spectrum of CDOM in the ultraviolet-visible band exhibits an exponential decay as the wavelength increases; $a$(350) is usually used to indicate the CDOM concentration and biogeochemical cycle process in lakes and rivers [11,12], so we select a(350) [33] as the parameter referring to the CDOM concentration.

Parameters associated with the absorption properties of chromophoric dissolved organic matter (CDOM) in water are frequently employed to indirectly assess the distribution, origin, composition, fate, and attributes of CDOM involved in geochemical cycles. The parameter SUVA$_{254}$, which represents the ratio between the absorption coefficient of ultra-

violet light at a wavelength of 254 nm and the concentration of dissolved organic carbon (DOC), primarily indicates the presence of aromatic compounds [34]. The 3D-EEMs was measured with a Hitachi F-7000 fluorescence spectrometer and the HIX was calculated using MATLAB [35]. Other water quality parameters such as chlorophyll (Chla) [36], total phosphorus (TP), total nitrogen (TN) [37], chemical oxygen demand ($COD_{Mn}$) [38], turbidity (Turb) [26], dissolved organic carbon (DOC) [39], total suspended solids (TSM) concentration [40], etc., were obtained by laboratory analysis and testing.

*2.4. Remote Sensing Data and Preprocessing*

The remote sensing data of Lake Khanka from 2016 to 2022 was obtained from the Sentinel3-OLCI sensor. These data were sourced from the Copernicus Data Center, which can be accessed at https://scihub.copernicus.eu/ (accessed on 1 January 2020). The optimized design of the OLCI incorporates 21 spectral channels that cover the visible to near-infrared spectral range. This design enables a more comprehensive and accurate representation of the spectral characteristics of complex water bodies. Partial atmospheric correction is performed using SeaDAS8.2.0 (https://seadas.gsfc.nasa.gov/) (accessed on 1 January 2020). There is currently no reliable atmospheric correction that can produce accurate OLCI Rrs data for Lake Khanka. Therefore, a partial atmospheric correction to correct for the gaseous absorption and Rayleigh (molecular) scattering effects was applied to the Level-1B data using routines and look up tables (LUTs) embedded in the software (SeaDAS, version 8.2.0), where Equation (2) was used to calculate the Rayleigh-corrected reflectance $R_{rc}(\lambda)$:

$$R_{rc}(\lambda) = \pi / F_0(\lambda) / cos\theta \times (L_t(\lambda)/t_{g_{sol}}(\lambda)/t_{g_{sen}}(\lambda) - L_r(\lambda) - TL_g(\lambda)) \tag{2}$$

where $L_t(\lambda)$ represents the irradiance of the top of the atmosphere (TOA), $L_r(\lambda)$ represents the ray radiance, $TL_g(\lambda)$ represents the flash radiance of TOA, $\theta$ represents the solar zenith angle, $F_0(\lambda)$ represents the average solar flux, $t_{g_{sol}}(\lambda)$ represents the gaseous transmittance from the sun to the surface, and $t_{g_{sen}}(\lambda)$ represents gaseous transmittance to the sun in Equation (2).

A total of 70 non-cloud images should be acquired within a one-week time frame or on the same day, aligning with the five designated sampling times between May and October. The software ArcGIS 10.7 is utilized to extract the Rayleigh-corrected reflectance of the image at the specified sampling point, denoted as $R_{rc}(\lambda)$.

*2.5. Model Construction and Evaluation*

The final inversion model utilized a selection of 17 bands, specifically Oa1 to Oa17, covering a wavelength range of 400 to 865 nm as the input variables. After the model was trained and verified successfully, error analysis was carried out, and the result prediction and batch inversion were largely completed. The correlation analysis of Sentinel-3 remote sensing reflectance for each band and *a*(350) absorption coefficient was conducted using matlab2016 software programming. This analysis includes various approaches such as single band analysis, band ratio analysis, and other band combinations. The objective is to develop a linear model that is appropriate for the Lake Khanka CDOM absorption inversion regression model. MATLAB (2016) software trained the correlation between single band, band ratio, and other band combinations of 17 OLCI $R_{rc}(\lambda)$ randomly and the absorption coefficient of CDOM. The highest correlation coefficient R is recorded in the addition of each band combination.

In order to determine the most suitable model, we conducted a comparative analysis of five machine-learning techniques. These techniques were applied to Sentinel-3 simulated reflectance data to retrieve CDOM. The following are examples of popular machine learning algorithms: 1. Gradient Boosting Decision Tree (GBDT); 2. eXtreme Gradient Boosting (XGBoost); 3. Support Vector Regression (SVR); 4. Back Propagation Neural Network (BPNN); 5. Random Forest (RF). The Random Forest Regressor package in Python works to build a Gradient Boosting Decision Tree model. The number of decision trees

affects the complexity and fitting performance of the model to a certain extent; thus, the hyperparameters are used to obtain the optimal parameter combination of the model. The independent variable utilized in the OLCI data reflectance is the input from bands B1-B17, while the dependent variable is the CDOM absorption $a(350)$ at a wavelength of 350 nm. Two thirds of the data are used as the modeling dataset, and the remaining one third is used as the verification dataset. The established linear regression model and machine learning model were compared and evaluated using R, RMSE, MAE, and other indicators.

$$RMSE = \sqrt{\frac{\sum_{i=1}^{n}(X_{estimated} - X_{observed})^2}{n}} \qquad (3)$$

$$MAE = \frac{1}{n}\sum_{n=1}^{n}|(X_{estimated} - X_{observed})| \qquad (4)$$

## 3. Results

### 3.1. Characteristics of CDOM and Water Quality Parameters

The concentration of CDOM is characterized by the absorption coefficient at 350 nm. The average value of the overall absorption coefficient of Lake Khanka is $4.69 \pm 2.28$ m$^{-1}$. The value is $4.77 \pm 1.26$ m$^{-1}$ in the Small Khanka, while it is $2.99 \pm 1.14$ m$^{-1}$ in the Great Khanka. The SUVA$_{254}$ in Lake Khanka is $4.99 \pm 1.38$, which is $4.85 \pm 0.77$ in the Small Khanka and $5.51 \pm 1.82$ in the Great Khanka. The overall range of water sample HIX obtained from five field experiments from 2017 to 2021 is $2.16 \pm 1.41$, including $2.39 \pm 1.59$ for the Small Khanka and $1.67 \pm 0.79$ for the Great Khanka. The concentration of Chla in Lake Khanka water was measured five times between 2016 and 2021. The average concentration was found to be $10.36 \pm 9.29 \mu$m/L. Similarly, the total nitrogen (TN) concentration was measured to be $0.57 \pm 0.21$ mg/L, total phosphorus (TP) concentration was $0.14 \pm 0.73$ mg/L, chemical oxygen demand (COD$_{Mn}$) was $3.30 \pm 0.19$ mg/L, total suspended matter (TSM) was $92.69 \pm 48.09$ mg/L, turbidity was $68.16 \pm 25.16$ NTU, and DOC concentration was $5.08 \pm 1.41$ mg/L.

### 3.2. Correlation between Lake Environmental Hydro-Chemical Characteristics and DOM

The relationship between parameters associated with CDOM in inland water and DOC is widely acknowledged and supported by substantial evidence, as depicted in Figure 2 [14,17,41–44]. The indirect inversion of the DOC concentration using the absorption properties of the optically active substance CDOM in water mainly relies on the conservative mixing relationship between CDOM and DOC. The data collected during five instances of field sampling in Lake Khanka between 2017 and 2021 revealed a noteworthy linear correlation between the absorption coefficient of CDOM at a wavelength of 350 nm and the concentration of DOC. The correlation coefficient value was determined to be 0.823 ($p < 0.01$), indicating a strong relationship. This finding suggests that the absorption coefficient $a(350)$ of CDOM in Lake Khanka can serve as a reliable indicator of DOC. Consequently, we have opted to utilize the absorption coefficient of $a(350)$ in conjunction with the measured DOC concentration to develop an empirical model for estimating the distribution of DOC concentration in Lake Khanka. In addition, the results of HIX show a strong correlation between HIX and CDOM (R = 0.78, $p < 0.01$), and CDOM absorption can be used as an indirect inversion of the spatial distribution of HIX.

The correlation analysis of water quality parameters involved the calculation of the Spearman correlation coefficient and the corresponding significance test $p$ value for each pair of water quality parameters. In addition to DOC, positive correlation factors with $a(350)$ include COD, SDD (Secchi-Disk Depth), Chla, pH, and salinity, and the linear correlation coefficient with COD is as high as 0.625 ($p < 0.01$). The CDOM inversion results reflect the temporal and spatial distribution and change trend of COD to a certain extent.

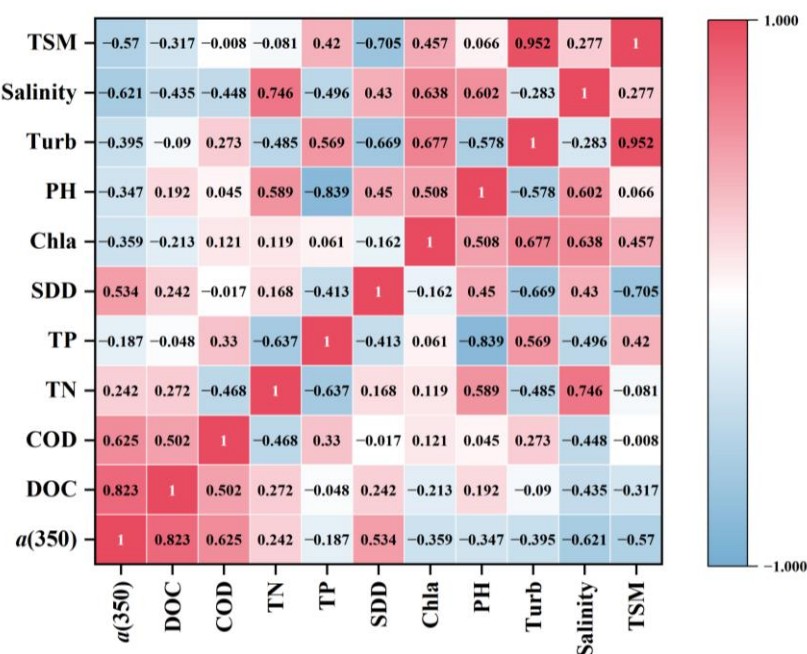

**Figure 2.** Correlation heat map of lake environmental water chemical parameters.

### 3.3. Modeling and Verification of CDOM and DOC

This study utilized two-thirds of the measured CDOM and the corresponding remote sensing reflectance data at sampling points to establish a concentration retrieving model for a wavelength of 350 nm. The remaining one third of the data were allocated to verify the accuracy of the model. We tried to use Matlab2016 and several machine learning algorithms to fit the relationship between the RrcB1-B17 band and CDOM absorption coefficient $a$(350). The initial step involved determining the appropriate model that correlates CDOM and single-band reflectance $R_{rc}(\lambda)$ within the spectral range of 400–865 nm, or considering the various combinations of each band. After a huge amount of training, including, but not limited to, what is shown in Table 1, we found that $[Rrc(412.5) + Rrc(490)]/[Rrc(400)/Rrc(490)]$ had the closest correlation with CDOM absorption (modeling R$^2$ = 0.38).

**Table 1.** Performance of different algorithms in Lake Khanka.

| Linear Model | Modeling R$^2$ | Verify R$^2$ |
|:---:|:---:|:---:|
| $a$(350) = −87.54b5 + 12.28 | 0.28 | 0.13 |
| $a$(350) = −82.74b4 + 11.38 | 0.27 | 0.13 |
| $a$(350) = −90.71b6 + 14.06 | 0.27 | 0.11 |
| $a$(350) = −45.71(b4 + b6) + 13.14 | 0.29 | 0.13 |
| $a$(350) = −64.01 (b6 − b13) + 10.71 | 0.18 | 0.08 |
| $a$(350) = −462.9(b4 × b6) + 8.60 | 0.29 | 0.14 |
| $a$(350) = −29.57(b15/b16) + 28.89 | 0.14 | 0.04 |
| $a$(350) = 52.34 [(b2 + b4)/(b1/b4)] + 14.56 | 0.38 | 0.20 |

Upon comparing the linear model (Table 1) with the machine learning model (Table 2), it has been observed that the machine learning model, constructed through nonlinear modeling using training data, exhibits superior performance and stability compared to the linear regression model. Consequently, machine learning has been chosen as the preferred approach for establishing the final model and implementing the inversion method.

**Table 2.** The accuracy evaluation of machine learning algorithm modeling and verification.

| ML Algorithms | Modeling $R^2$ | Verify $R^2$ | RMSE (m$^{-1}$) | MAE (m$^{-1}$) |
|---|---|---|---|---|
| SVR | 0.47 | 0.43 | 1.05 | 0.73 |
| BP | 0.68 | 0.59 | 0.81 | 0.55 |
| XGBoost | 0.74 | 0.56 | 0.73 | 0.53 |
| RF | 0.82 | 0.72 | 0.60 | 0.42 |
| GBDT | 0.95 | 0.84 | 0.31 | 0.23 |

The best fitting model is created by GBDT (modeling $R^2$ = 0.95) which is better than that of the band combination linear model produced by Matlab2016 and any other machine learning algorithms we have ever tried. Consequently, in this study, the machine learning approach was employed to model and retrieve the CDOM concentration of Lake Khanka (as shown in Figures 3–7).

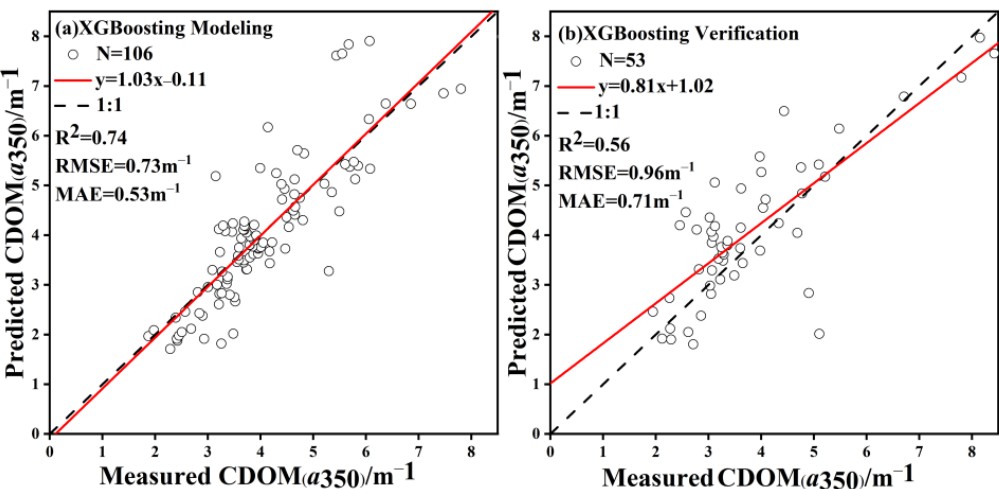

**Figure 3.** Modeling and validating scatterplots of eXtreme Gradient Boosting.

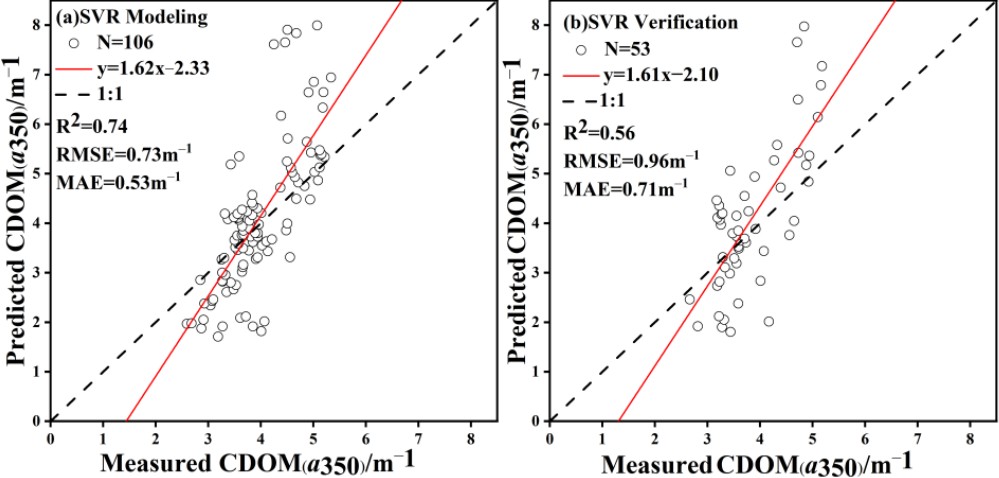

**Figure 4.** Modeling and validating scatterplots of Support Vector Regression.

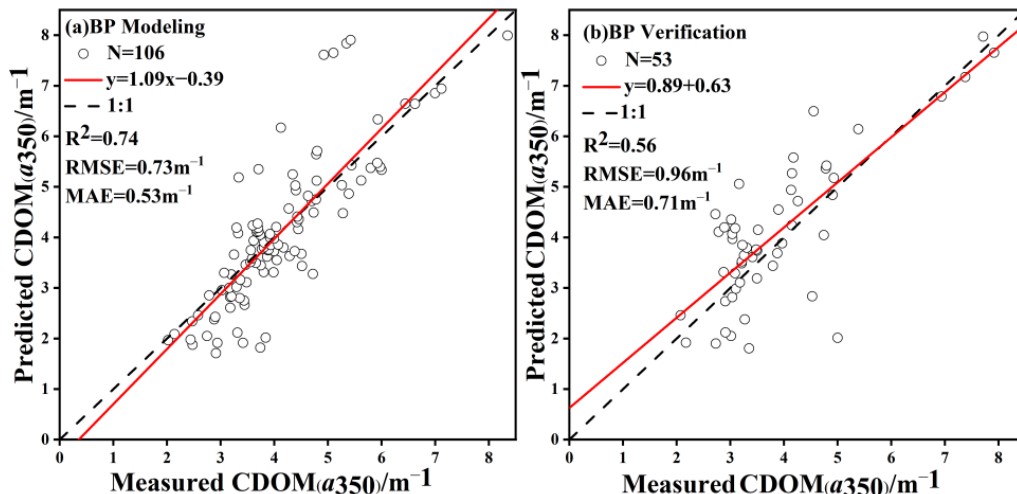

**Figure 5.** Modeling and validating scatterplots of Back Propagation.

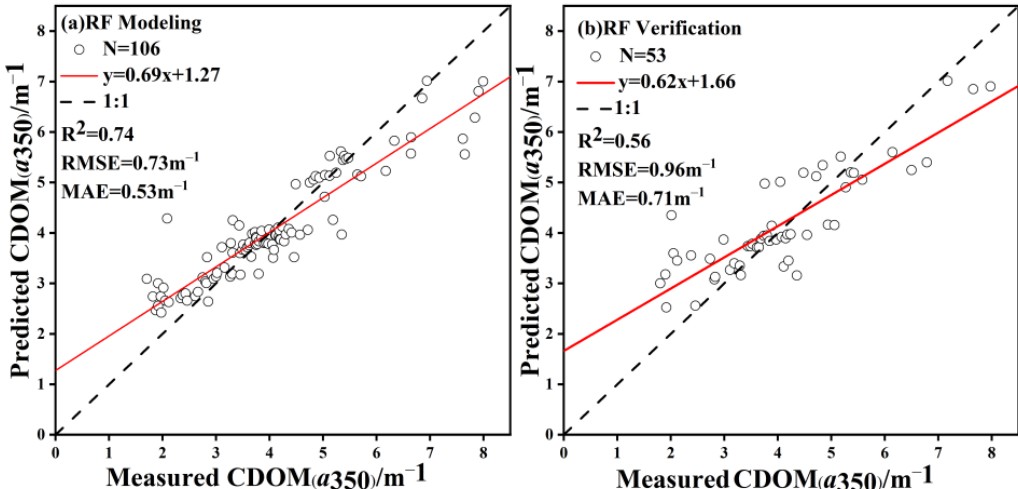

**Figure 6.** Modeling and validating scatterplots of Random Forest.

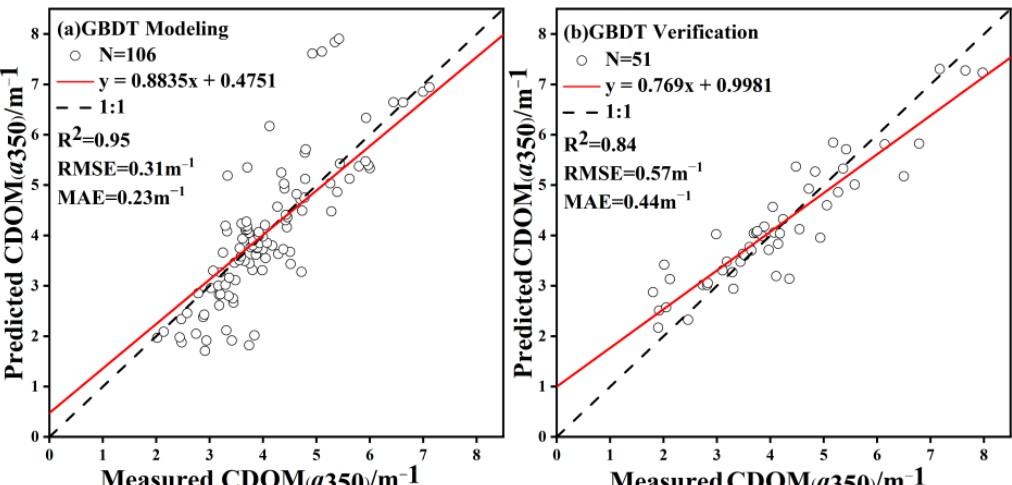

**Figure 7.** Modeling and validating scatterplots of Gradient Boosting Decision Tree.

Based on the analysis mentioned in Section 3.2, a prediction model for DOC concentration was established using two thirds of the measured CDOM absorption and DOC concentration data selected randomly. The accuracy of the model was then verified using

the remaining one third of the data. The independent variable selected for establishing the linear regression equation between CDOM and DOC was the index with the highest correlation within the absorption parameters of CDOM. This selection was made using SPSS (version 26.0). The model accuracy is shown in Figure 8. $R^2 = 0.69$, indicated that the CDOM variable is the standard measure of DOC.

$$DOC = 0.7779aCDOM(350) + 1.8968 \qquad (5)$$

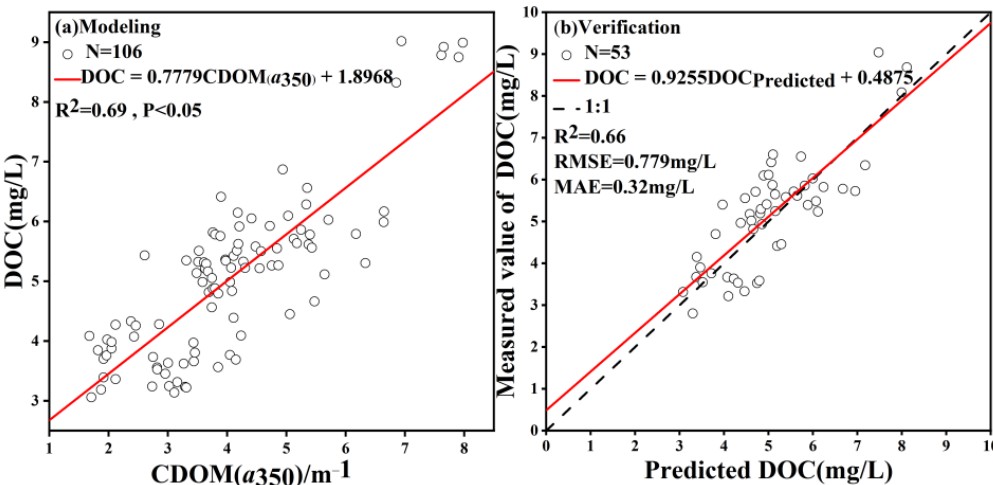

**Figure 8.** Construction and verification of CDOM absorption coefficient and DOC concentration model.

### 3.4. The Distribution of DOC for Lake Khanka

Our efforts were focused on the selection of cloudless or minimally cloudy images of Sentinel-3 from the period of 2016 to 2022. Subsequently, we applied the GBDT model established earlier to invert the CDOM value of a single image. The inversion results are integrated into the spatial distribution map representing the annual mean value. Then, according to the good linear correlation between CDOM-DOC, the grid calculator was used to enter the model formula to obtain the spatial distribution of the annual average DOC concentration in 2016–2022. The results are presented in Figure 9. The annual mean value of CDOM absorption coefficient for Lake Khanka falls within the range of 3.26–4.09 m$^{-1}$. Additionally, the annual mean value of DOC ranges from 4.56 to 5.19 mg/L.

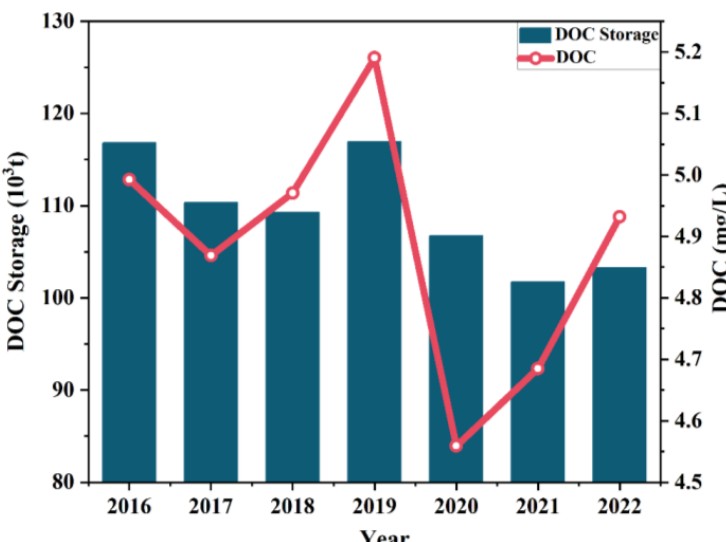

**Figure 9.** Annual mean value fluctuation and reserves of DOC in lake Khanka during 2016–2022.

The interannual variation in DOC concentration in Lake Khanka is minimal, and the overall level remains consistent over time. The provided illustration demonstrates the fluctuation of DOC value over a specific period. From 2016 to 2019, the value of DOC experienced an initial increase. However, in the subsequent period of 2019 to 2021, the value declined. Eventually, in 2022, the value stabilized at an average of 4.89 mg/L. Simultaneously, we estimated the DOC reserves of the lake to be $109.31 \pm 5.56$ ($10^3$ t). The normal DOC reserves [45] of Lake Khanka shown in Figure 10 are consistent with the annual DOC concentration changes. We also established a regression model ($R^2 = 0.61$, RMSE = 0.88) based on good correlation between CDOM and HIX to inverse the spatial distribution and mean of HIX. The linear model can be found in the Supplementary Materials.

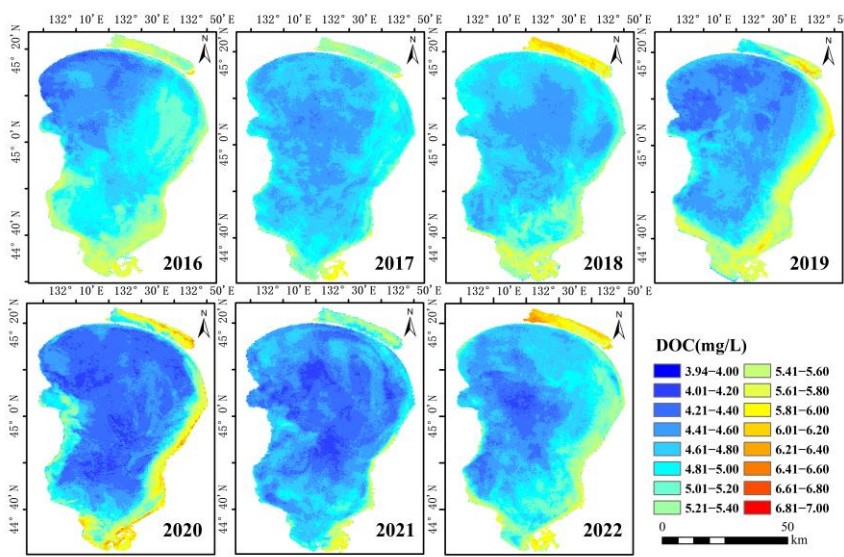

**Figure 10.** Inversion results of DOC concentration distribution in lake Khanka during 2016–2022.

From the spatial dimension (Figure 10), according to the DOC concentration inversion results, the overall trend is that the value of DOC in Small Khanka is bigger. The occurrence of low values was observed exclusively in Great Khanka, indicating that the overall water quality of Great Khanka is comparatively superior to that of Small Khanka. The concentration of DOC in the northwest of Small Khanka was tiny, while that in the southeast was great.

## 4. Discussion

### 4.1. Advantages and Disadvantages of the Model

The GBDT algorithm in machine learning is a classification prediction technique that relies on decision trees. It has the capability to combine multiple tree models and minimize errors through the gradient boosting process [46]. CDOM represents a fraction of the overall organic matter present in water, while DOC serves as a measure of the concentration of organic matter by quantifying its carbon content. CDOM is considered to be subordinate to DOC in its nature. Therefore, the indirect inversion of the temporal and spatial distribution of DOC through the relationship between CDOM and DOC has a theoretical basis.

When considering error estimation, it is important to note that various atmospheric correction algorithms have distinct impacts on OLCI remote sensing images [47]. The Rayleigh scattering correction we use may not be able to eliminate the possible adjacency effect absolutely on the coastal area of Lake Khanka. Due to the signal-to-noise ratio in the NIR bands, eventual residual NIR contributions from the bottom resuspension, and anisotropic reflectance of the land surface may influence adjacency effects in the NIR [48]. Unfortunately, an accurate estimate and likely correction of adjacency effects relies on the determination of the actual land albedo at the same time, and, at the same bands, our atmospheric correction cannot overcome the low SNR of the OLCI sensor. Future missions

targeted at the observation of inland waters should take this issue into account. In the long run, improved atmospheric correction is still required to obtain reliable *Rrs* data under both optimal and non-optimal observing conditions, as most algorithms do rely on *Rrs* data. In the GBDT modeling process, the input bands involved in the calculation have a limited ability to eliminate the influence of the atmosphere on the image quality; thus, the original input variables may lead to some inevitable deviations to the real values of the images and bands. Additionally, the optical complexity of inland water is commonly influenced by elevated levels of phytoplankton biomass, Chla, non-algae particulate matter, and other factors. This overlapping absorption poses challenges in discerning the signals of multiple optically active substances during remote sensing [49–51]. One additional aspect to consider is the impact of CDOM and total suspended matter (TSM) on the attenuation of light penetration in water, which subsequently affects the phenomenon known as the water adjacency effect [50,52,53]. In that, high concentrations of TSM can cause an overestimation of CDOM, which should be considered to improve atmospheric correction in Case 2 water and reduce the signals of other optical components in NIR by employing a novel band ratio combination. Last but not least, in recent years, most algorithms for estimating CDOM through remote sensing technology have been empirical or semi-empirical [17,22,54], lacking the theoretical basis of radiative transfer, and the extension of the model is contingent upon the limitations imposed by specific regional conditions, as determined through measurements and experiments.

### 4.2. Factors Affecting the Change in DOM

CDOM-DOC exhibits significant heterogeneity on both temporal and spatial scales. Its dynamic fluctuations are primarily influenced by the combined impacts of diverse human activities, including agricultural runoff, industrial pollution, and natural factors within the watershed. These natural factors encompass solar radiation, temperature (TEMP), precipitation (PRCP), Wind Speed (WDSP), Station Pressure (STP), as well as disturbances, e.g., wastewater discharge, material transport, and energy conversion in water and sediments caused by human activities.

#### 4.2.1. Natural Environmental Attributions

We collected the daily and monthly natural factors at the meteorological station in JiXi with DOC concentration for multiple regression analysis (Figure 11). Multiple regression analysis was performed in SPSS on the natural factors (TEMP, STP, WDSP, and PRCP) corresponding to every single date. In the multiple regression model, Beta represents the standardized regression coefficient, representing the degree to which the dependent variable can be explained. Thus, Beta represents the degree of impact of such factors on DOC. From the perspective of distinct time scales, TEMP has the greatest impact within a day, followed by PRCP, WDSP, and STP; but, within a month, the influencing factors in descending order are STP, TEMP, and WDSP. The long-term influence of various natural factors results in a consistent effect on the concentration of DOC. Specifically, the summer season is typically characterized by high temperatures and abundant rainfall in the Xingkai–Muleng basin. This climatic pattern results in a significant influx of organic matter from the upstream river, leading to an elevated concentration of organic pollutants. Over one day, temperature and precipitation have the greatest impact. The phenomenon of high-intensity precipitation induces the soil-leaching effect, resulting in an influx of water supply to the lake and a rapid increase in runoff into the lake, which dilute the concentration of organic pollutants in the current lake, accelerate the metabolism of water and also the reproduction and death of microorganisms, consume a large amount of organic matter, and, as a result, reduce the concentration of CDOM, DOC, and COD. But, within 30 days, the combination of temperature, pressure, and wind speed can explain a greater impact. The temperature can affect the decomposition rate of organic matter. Since Lake Khanka is relatively shallow (1.70–3.99 m), the action of strong winds

and atmospheric pressure on the underlying surface will cause sediment resuspension, which may release a large amount of organic matter.

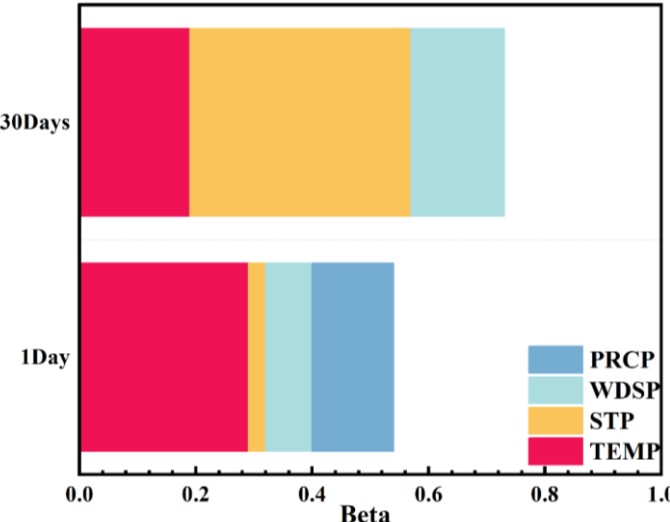

**Figure 11.** Contribution rate of natural factors (PRCP, WDSP, STP, and TEMP) to DOC changes in 1 day and 30 days in multiple regression.

Considering the hydrological conditions, the Muleng River, one of the main tributaries of the Ussuri River, flows through Jixi City (a coal and industrial city with a population of 1.6 million) with tremendous DOC which is severely polluted by industrial wastewater and sewage. Songacha River, the only discharge river to the east of Lake Khanka, has a minimum water depth of only 1 m, and a natural overflow weir has been formed at the mouth of the river, which seriously hinders the discharge of the lake area, causing the lake in the east of Lake Khanka to inherit more organic matter. From the perspective of topography, the topography DEM of the Muleng–Xingkai watershed where Lake Khanka is located is generally higher in the northwest (250–500 m) and lower in the southeast (50–72.5 m), which results in the flow of the Lake Khanka system from west to east. However, these natural factors cannot explain 100% of the changes that might happen in DOC, and other natural and human factors require profound exploration.

### 4.2.2. Human Activity Attributions

Due to the deficiency of accurate data on interannual changes in human factors from 2016 to 2022, they are insufficient to support data analysis. However, we can still deconstruct the impact of human activities on CDOM, DOC, and the water environment from human traces in land use patterns. The land use on the west bank of Lake Khanka is mainly cultivated land and artificial surface. The excessive application of agricultural chemical fertilizers, as well as the presence of aquaculture and livestock breeding in the vicinity, result in agricultural runoff. This runoff carries a significant quantity of organic matter into Lake Khanka, leading to the enrichment of various nutrients and organic pollutants in the lake. The east bank of Lake Khanka is dominated by large wetland grasslands, and the surrounding wetland ecosystem has become the main source of DOC on the east bank of Lake Khanka because of its rich organic matter and huge carbon storage capacity. This is consistent with the results of the research of Liu [55]. Notably, 2020 was the first year of the new coronavirus outbreak. The epidemic has led to a reduction in tourism and agricultural activities, and the average annual DOC concentration has dropped sharply compared to 2019. In 2021, this new epidemic appeared to be easing, and the DOC concentration gradually increased as production and life resumed. HIX is almost consistent with CDOM and DOC changes (Figure 12). The weak HIX (HIX < 4) indicated that Lake Khanka was abundant with more biological or recent autochthonous components [35] and that the large abnormal fluctuations (HIX > 4) might be caused by interference from human

activities. Consequently, this process directly or indirectly contributes to the conversion of these substances into CDOM or DOC, among other compounds. Additionally, SUVA provides insights into the level of humification of organic matter in water; it is shown to be a useful parameter for estimating the dissolved aromatic carbon content in aquatic systems [56]. $SUVA_{254}$ of the Great Khanka is greater than that of the Small Khanka, and the large standard deviation indicates that the Great Khanka is more affected by terrestrial aromatic organic carbon input.

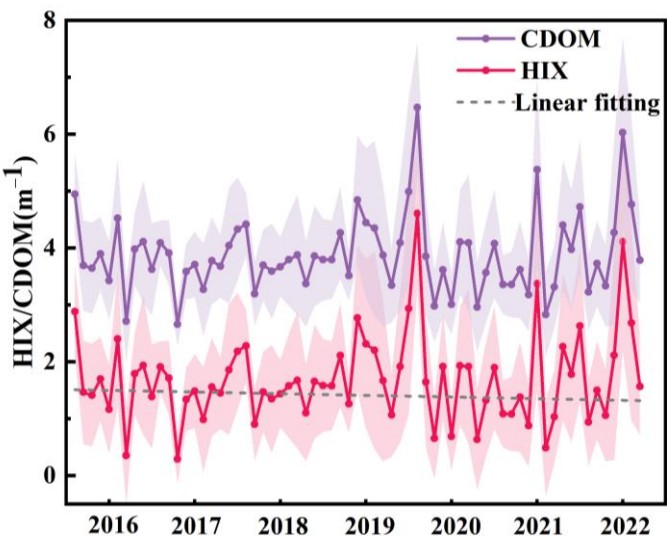

**Figure 12.** CDOM and HIX average values in the inversion results of Lake Khanka images in chronological order.

### 4.3. Environmental Significance and Prospect of Lake Potential Water Quality Parameter Evaluation

4.3.1. Implications for Potential Water Quality Parameters and Carbon Cycling in Lakes

The water quality of Lake Khanka significantly influences the economic development and ecological equilibrium between China and Russia. The measurement of total nitrogen, total phosphorus, and permanganate index COD are crucial water quality parameters for assessing the level of eutrophication of lakes. Because these indicators do not have optical properties, the indirect inversion of these substances by remote sensing is used to infer their concentrations. A notable correlation exists between the primary indicators of organic pollution, namely COD and CDOM absorption coefficient, and the concentration of DOC in the Great Khanka and Small Khanka water bodies. The estimation of CDOM-DOC by means of remote sensing can further estimate the concentration and distribution of COD indirectly, and then conduct more comprehensive testing and analysis of water quality conditions. The correlation between COD and DOC exhibits a stronger relationship (R = 0.78, $p < 0.05$). Consequently, it is crucial to prioritize the management of DOC in efforts aimed at mitigating the migration and transformation of pollutants in the water environment.

The application of remote sensing technology for the large-scale inversion of DOC concentration continues to encounter significant challenges. Many studies [57–62] have shown that the current relationship between DOC concentration and carbon flux is unprecedented, and relevant hypotheses indicate that the continued rise in DOC levels will have unpredictable consequences for the global carbon cycle [63,64]. Long-term DOC increases may have wide-ranging impacts on freshwater biota, drinking water quality, and inland water ecosystems. More specifically, the agricultural and fishery benefits of the farms around Lake Khanka will be severely affected, and humans will face more risks regarding drinking water and food safety along with unprecedented threats to ecological balance.

### 4.3.2. Shortcomings and Prospects of the Study

CDOM exhibits a specific vertical divergence pattern within the aquatic environment. The concentration and composition of CDOM in the surface layer of water frequently exhibit significant disparities compared to the deep layer. This discrepancy arises primarily from the process of photochemical degradation. The optical signals detected by remote sensing sensors in deep water are strongly correlated with water quality parameters, specifically water transparency and turbidity. Excluding the sensor's sensitivity to water optical signals, Lake Khanka has low transparency ($21.50 \pm 5.89$ cm) and high turbidity ($68.16 \pm 25.16$ NTU). The main optical signal of the water body comes from optically active substances such as surface CDOM; however, the concentration and distribution of substances in the lower layer have little effect on the optical response of the surface. In this particular scenario, it is recommended that further investigation be directed towards the longitudinal distribution of CDOM in aquatic environments. When employing CDOM as an indirect means to assess the lateral spatial distribution of DOC, this variability introduces inconsistencies in CDOM measurements.

Since the latitude of Lake Khanka varies from $43°55'$ to $46°30'$, the lake is located in the north temperate monsoon climate zone where the winter is cold and lasts long, and thus the freezing period is about 168 days [65]. The presence of a thick ice layer hinders the remote sensing inversion of CDOM. Consequently, the temporal and spatial distribution of CDOM in Lake Khanka may be overestimated if the annual average is not taken into account and glacial image data are excluded. More attention should be paid to considering these conditions in the future monitoring of CDOM and DOC.

The OLCI data possess a significant time resolution that enables efficient utilization of both macro and micro time scales for the purpose of monitoring lake water quality. In the future, the integration of daily, monthly, and annual variations in water parameters will be employed to assess alterations in lake water quality and biogeochemical cycles. The limited duration of Sentinel-3's launch, spanning from 2016 to present, has resulted in an insufficient time frame for the available annual remote sensing data. Additionally, there is an inadequate quantity of matched driving factor data to support comprehensive data analysis. As a result, the further exploration of natural factors and human activities that have influenced the spatial and temporal variation in CDOM-DOC should be implemented properly.

## 5. Conclusions

Results from the research indicate that the modeling approach utilizing machine learning GBDT outperforms the linear regression model based on band combination in accurately predicting the concentration of CDOM in Lake Khanka from 2016 to 2022. A comprehensive analysis was conducted on a monthly basis from 2016 to 2021, specifically focusing on the nonglacial period. The parameters examined encompassed both optical and non-optical characteristics of inland water, providing an accurate representation of the water quality conditions. The analysis of the spatial distribution of CDOM and DOC reveals that the Small Khanka Lake exhibits a larger size compared to the Great Khanka Lake. Additionally, the perimeter of the lake is generally greater than its central region. The concentration of DOC in the Great Khanka is reported to be $4.74 \pm 0.12$ mg/L, while in the Small Khanka it is $5.50 \pm 0.30$ mg/L. These measurements were recorded between the years 2016 and 2022.

The establishment of the CDOM-DOC relationship enables the potential prediction of DOM distribution and sources. Additionally, the correlation coefficient with COD and HIX also further indicates the occurrence of organic matter in water. These variations in size and distribution may be attributed to the impact of natural factors, human industrial and agricultural activities along the coast, as well as the succession of the aquatic ecological environment. Meteorological factors including temperature, wind speed, precipitation, and pressure, play a role in the distribution of CDOM in the short term. In order to ensure comprehensive planning and sustainable management, it is imperative to

prioritize the examination of long-term land use alterations and potential impacts on the water environment resulting from human activities. Given the significant geographical location, ecological functions, and economic value attributed to Lake Khanka, ensuring the effective monitoring and treatment of water quality within its watershed is crucial for promoting sustainable development for both human and natural systems. This study provides valuable insights into the analysis of organic matter in lakes and the distribution of carbon sources. Additionally, it establishes a strong foundation for future investigations into the source and composition of carbon in lakes.

**Supplementary Materials:** The following supporting information can be downloaded at https://www.mdpi.com/article/10.3390/rs15245707/s1, Figure S1. CDOM absorption coefficient curve of samples from 200 to 800 nm. Figure S2. Sentinel 3 OLCI reflectance *Rrc* after atmospheric correction between Oa1–Oa16. Figure S3. Accuracy verification of HIX and CDOM concentration model. Table S1. Sentinel-3 Ocean and Land Color Instrument (S3-OLCI) spectral and spatial specifications. Table S2. Descriptive statistics of water quality parameters.

**Author Contributions:** S.Q.: Conceptualization, data curation, formal analysis, methodology, visualization, roles/writing—original draft. K.S.: funding acquisition, project administration, resources. Y.S.: funding acquisition, project administration, supervision, writing—review and editing. F.L.: software, investigation, resources. Z.W.: investigation, resources. G.L.: software, resources. H.T.: investigation, resources. Y.L.: funding acquisition, project administration, investigation. All authors have read and agreed to the published version of the manuscript.

**Funding:** This research was funded by the Strategic Priority Research Program of the Chinese Academy of Sciences (XDA28100100), Natural Science Foundation of China (No. 41231390 and NO.41971322), the Science and Technology development plan project of Jilin Province (No. 20220508017RC), the Special Postdoctoral Fellowship of Jilin Province of China granted for Dr. Yingxin Shang, the Youth Innovation Promotion Association of Chinese Academy of Sciences (2020234), Young Scientist Group Project of Northeast Institute of Geography and Agroecology, Chinese Academy of Sciences (2022QNXZ03), Municipal Academy of Science and Technology Innovation Cooperation Project, Changchun Surface Water Quality Sky-Ground Integrated Remote Sensing Monitoring Technology Research and Development (21SH10), and Jilin Province and Chinese Academy of Sciences Science and Technology Cooperation High-Tech Industrialization Special Fund Project (2021SYHZ0002).

**Data Availability Statement:** Data are contained within the article and Supplementary Materials.

**Conflicts of Interest:** The authors declare no conflict of interest.

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
