# Peer review of "Remote Sensing Estimation of CDOM and DOC with the Environmental Implications for Lake Khanka"

_remotesensing, doi:10.3390/rs15245707_

Round 1
Reviewer 1 Report
Comments and Suggestions for Authors
I have carefully reviewed your manuscript and find that it demonstrates a well-structured framework, comprehensive results, and other commendable qualities. I recommend accepting the manuscript for publication.
Reviewer 2 Report
Comments and Suggestions for Authors
The article concerns the important problem of remote assessment of the content of dissolved organic matter and dissolved organic carbon in the water of inland water bodies. The model proposed by the authors is a necessary step in the environmental monitoring of eutrophic waters.
Unfortunately, the article is not without its shortcomings, especially numerous lexical errors that make it difficult to understand the essence of the work.
Only 7 out of 67 cited references are recent (within the last 5 years), which does not meet the formal requirements.
The approach used by the authors raises some doubts. Firstly, as far as I could understand, the authors perform atmospheric correction of satellite data only accounting for the Rayleigh scattering component and glints. Thus, the relative variation in upwelling radiance at TOA caused by changes in CDOM concentrations will be significantly less than when corrected for the aerosol component. Additionally, CDOM variability affects the same short-wavelength part of the spectrum as atmospheric variability. Particularly, weak correlation between CDOM absorption and spectral combinations of Rrs prove that.
In Section 4.1 the authors mention the possible influence of the atmospheric correction, but make no attampt to evaluate its effect on the obtained models.
The conclusions are consistent with the presented data, but their scientific significance is not emphasized. In the Conclusions section, it is advisable to present not only generalized research results, but also specific ones that are of the greatest importance.
The text discusses not only CDOM, but also DOC, so it is recommended to include it in the paper title.
Specific comments:
Line 17: "DOC" require explanation, like you did with CDOM above.
Line 148: "a mixed fiber filter membrane (peninsula 0.45µm)"... Please check the whole manuscript for the incorrect wording. I happen to know a bit of Mandarin, and I keep wondering how did the automatic translation turn "pore diameter" into "peninsula". A miracle indeed.
L 190: Reference 44 does not refer to SeaDAS system. May be you wanted to refer to a certain atmospheric correction algorithm?
Eq 3: Why did you not substract the aerosol component from the TOA radiance? Please explain your approach in more detail.
L225: Sentinel3 band B1 wavelength is 400nm, and others are even bigger. You are trying to connect absorption at 350 nm with Rrs at these bands. If you have absorption measurements in the whole visible range, why not choose, for example, a(400) as a reference parameter for CDOM? It is possible that connection between Rrs at 400 nm and absorption at 350 could be too weak, considering CDOM variability and 50nm gap. Please explain the choice of reference wavelength for the absorption.
L233-245: " The concentration of Chla in Lake Khanka water was measured five times between 2016 and 2021."
Are all these average parameters (Chla, TN, TP etc) obtained on the same 170 samples that you have mentioned in Section 2.2? Or was it 5 independent samples?
L266: What does SDD stand for?
Figures 3-7 present in total 106+53 = 159 data points. While the Table S2 in the supplementary material shows 151 data points for CDOM.
L388: How did you calculate Beta? How exactly does it represent the degree of impact?
Too many lexical errors due to the automatic translation. Please check the proper choice of words.
Reviewer 3 Report
Comments and Suggestions for Authors
Dear authors
I find your paper very interesting, the data-set is robust, the results obtained is very good and the structure of the paper is well done.
I think only problem of the paper could be related to the robustness of atmospheric corrections. In particular I find the reflectances very high in the NIR region, this could be due to the presence of adiacency effects in the coastal area of the the lakes. The problem of adiacency is clear present in the maps where the values is to higher. I think the authors must discuss about this point in the discussion if is not able to apply some correction of adiacency.
Another point is related to the overestimation of the CDOM in presence of high values. This could be due to the high concentration of TSM that influence the spectral magnitude and shape of spectral signature. I think the authors must add a discussion related to this point
Round 2
Reviewer 2 Report
Comments and Suggestions for Authors
L69-71 "HIX represents the degree of humification of DOM, while HIX semi-quantitatively represents the possible duration of organic matter in the environment."
There is something wrong with this phrase, please check.
L106 - water color, not watercolor (it is a kind of paint).
Fig8 still has 100+51 = 151 data points for DOC and CDOM, while in the Table S2 you have corrected DOC data amount to 159.
Thank you for the explanation of Beta value. I suggest you add it to the paper.
In general, I think that this manuscript has been significantly improved, and after correcting the abovementioned minor flaws, it is suitable for publication.
As far as I can see, now the English of this manuscript is good enough.
